# Influence of sudden stratospheric warming with elevated stratopause on the hydroxyl in the polar middle atmosphere

**Jin Hu, Sheng-Yang Gu, Yusong Qin, Yuxuan Liu, and Yafei Wei**

School of Earth and Space Science and Technology, Wuhan University, Wuhan, China

**Correspondence:** Sheng-Yang Gu (gushengyang@whu.edu.cn) and Yusong Qin (qinyusong@whu.edu.cn)

**Abstract.** Based on the specified dynamics simulation of Whole Atmosphere Community Climate Model with ionosphere/thermosphere extension (SD-WACCM-X), the composite response of polar hydroxyl radical (OH) layer in the mesosphere and lower thermosphere (MLT) to the Arctic sudden stratospheric warming (SSW) events with elevated stratopause (ES) during 2004–2023 is investigated. A total of ten ES-SSW events are systematically analyzed. Before the onset of ES-SSW events, the OH concentration climatologically peaks at $\sim 7.4$ ppbv near 82.4 km. During the stratospheric warming phase, relative to the climatology, the peak height of OH layer undergoes a distinct upward displacement reaching $\sim 85.9$ km accompanied by a reduction in the OH concentration to $\sim 2.9$ ppbv. This shift is closely linked to an $\sim 11\%$ and $\sim 90.8\%$ reduction in mesospheric temperature and atomic oxygen, respectively, due to enhanced upward residual circulation. During the elevated stratopause phase, the peak height of OH layer experiences a pronounced downward shift to $\sim 80.6$ km with a maximum in OH concentration to $\sim 6.8$ ppbv. This phase is characterized by $\sim 3.7\%$ and $\sim 137.3\%$ enhancements in mesospheric temperature and atomic oxygen concentrations, respectively, which is driven by intensified downward residual circulation. Further analysis suggests that OH concentration variations are positively correlated to mesospheric temperature anomalies and atomic oxygen redistribution induced by vertical transport, which is attributed to the significant influence of ES-SSW on gravity wave drag (GWs) in the mesosphere.

**Key points.** The peak height of Arctic OH layer rises during the stratospheric warming phase and descends during the elevated stratopause phase.

The change in OH concentration during ES-SSW shows a significant positive correlation with the change in mesospheric atomic oxygen concentration and temperature.

The vertical circulation anomalies due to the variations in gravity wave drag during ES-SSW alter the mesospheric atomic oxygen concentration and temperature.

## 1 Introduction

The middle atmosphere, spanning from $\sim 20$ to $\sim 100$ km in altitude, plays a vital role in coupling different atmospheric layers and modulating space weather phenomena. Hydroxyl radical (OH) is a key component in the MLT region, exerting an essential function in atmospheric chemistry and serving as an important indicator for assessing the atmospheric thermal budget. In the mesopause region, OH is primarily produced through the reaction between ozone and atomic hydrogen, forming excited OH* near 87 km. This excited state is deactivated either via photon emission in the Meinel bands (observed as nightglow) or by collisional quenching. The latter process prevails at lower altitudes, where higher atmospheric density facilitates the formation of a ground-state OH layer near 82 km (Damiani et al., 2010).

Despite the relatively well-understood chemical formation of OH, comprehensive observational characterization of its variability remains limited. Nevertheless, measurements from ground-based instruments and satellites, such as Thermosphere Ionosphere Mesosphere Energetics and Dynamics (TIMED)/Sounding of the Atmosphere using Broadband

Emission Radiometry (SABER) and Aura Microwave Limb Sounder (MLS), have provided a valuable opportunity to examine OH variability (Damiani et al., 2010; Medvedeva et al., 2019). Shapiro et al. (2012) identified a positive correlation between mesospheric OH variability and 27 d solar irradiance cycles, with stronger responses observed during periods of heightened solar activity. Minschwaner et al. (2011) further demonstrated that ultraviolet actinic flux serves as the primary driver of OH diurnal variability. Li et al. (2005) observed that OH concentrations peak near local noon, followed by a pronounced decline during nighttime. Additionally, statistical analyses by Gao et al. (2016, 2011, 2010) on the temporal and spatial distributions of OH and $O_2$ nightglow emissions highlighted significant local time variations, seasonal dependencies, hemispheric asymmetries, and a strong dependence on solar activity. Grygalashvyly et al. (2021) examined the semi-annual variation of the excited OH emission layer at mid-latitudes, attributing the results to the semi-annual oscillation of atomic oxygen and temperature.

Among the most prominent dynamical phenomena in the middle atmosphere are stratospheric sudden warmings (SSW), a large-scale wintertime phenomenon characterized by a rapid stratospheric temperature increase, zonal wind reversal, and substantial disruptions in atmospheric circulation (Manney et al., 2008; Chen et al., 2016; Bolaji et al., 2016; Gu et al., 2021). Studies have shown that SSW events are able to greatly modulate variations of atmospheric chemistry parameters (e.g., Kumar et al., 2024), particularly OH concentration (e.g., Winick et al., 2009). For example, Winick et al. (2009) attributed variations in OH layer to concurrent changes in temperature and atomic oxygen concentration in the upper mesosphere during the winters of 2004 and 2006. Similarly, Gao et al. (2011) documented reductions in OH and $O_2$ emissions associated with the January 2009 SSW, whereas Medvedeva et al. (2019) identified longitudinal disparities in OH emissions during the January 2013 SSW, which were likely modulated by variations in vertical wind patterns.

Elevated stratopause (ES) events refer to episodes during which the winter polar stratopause initially descends, subsequently becomes indistinct, and eventually reforms at a significantly higher altitude (Manney et al., 2008). Such events arise from strong forcing of the zonal wind and meridional circulation by planetary waves (de la Torre et al., 2012). In certain cases, ES events occur in connection with SSW events. According to Chandran et al. (2013), ES-SSW events are distinguished by a prolonged reversal of the stratospheric jet, enhanced gravity wave forcing, and intensified mean meridional circulation, relative to winters when SSWs occur without an accompanying ES. Compared with typical SSWs, ES-SSW events are particularly noteworthy because their enhanced downward transport significantly modifies the concentration of minor species in the MLT region (Chandran et al., 2013), such as OH. Given these persistent anomalies, fo-

cusing on ES-SSW events provides a physically meaningful basis for composite analysis. Nevertheless, the investigations into the impacts of ES-SSWs on polar mesospheric chemistry remain limited, owing to the sparse sampling of satellite observations, as exemplified by the TIMED/SABER satellite, which follows a precession orbit and requires several days to achieve full local time coverage. As a result, the influences of ES-SSWs on polar mesospheric OH concentration remain insufficiently understood.

Numerical simulations offer an alternative approach for investigating variations in the OH layer under dynamically complex conditions. The Specified Dynamics Whole Atmosphere Community Climate Model with ionosphere/thermosphere eXtension (SD-WACCM-X) offers a physically consistent representation of mesospheric chemistry and dynamics, allowing for a more continuous and spatially resolved analysis of OH layer responses to ES-SSW events. Previous studies have successfully employed SD-WACCM-X to investigate atmospheric dynamics procession (e.g., Zhang et al., 2025; Orsolini et al., 2022), which demonstrates that the SD-WACCM-X simulations provide a valuable opportunity for capturing OH variations during ES-SSWs in the absence of comprehensive observational coverage.

This study aims to explore the responses of OH concentrations to ES-SSW events using the SD-WACCM-X simulations for the period 2004–2023. Section 2 provides a brief introduction to the datasets, the definition of ES-SSW, and the analysis methods used in this study. Section 3 presents the results, focusing on the peak values and peak height of OH concentrations, along with their temporal evolution in response to ES-SSW events. Section 4 discusses the roles of mesospheric temperature and atomic oxygen in changes of OH layer and the underlying mechanisms of OH concentration variations during ES-SSW events. Finally, Sect. 5 TS1 summarizes the key findings and their implications for mesospheric dynamics during ES-SSWs.

## 2  Data and method

### 2.1  SD-WACCM-X

In this study, the Whole Atmosphere Community Climate Model with ionosphere/thermosphere eXtension (WACCM-X; Liu et al., 2018), an extended version of WACCM embedded within the Community Earth System Model version 2.2.0 (CESM2.2.0) framework developed by the National Center for Atmospheric Research (NCAR) (Gettelman et al., 2019; Danabasoglu et al., 2020), is employed to consider the characteristics of OH layer in the polar mesosphere. WACCM-X is an atmospheric model that simulates atmospheric processes from the surface ($\sim 0$ km) up to the ionosphere/thermosphere, extending to $\sim 700$ km depending on solar activity, which provides a detailed representation of dynamic, chemical, and radiative processes in the stratosphere, mesosphere, and thermosphere. The WACCM-X used in this study

is based on the Community Atmosphere Model-6 (CAM-6) physics and the three-dimensional chemical transport Model for Ozone and Related chemical Tracers (MOZART) chemistry (Gettelman et al., 2019). Specifically, WACCM-X incorporates nonorographic gravity wave drag parameterization, solar and geomagnetic forcing, and comprehensive gas and aerosol chemistry (Lee et al., 2021). The middle atmosphere scheme with the D-region chemistry (MAD) is used.

The specified dynamics (SD) configuration of WACCM-X is nudged by Modern-Era Retrospective Analysis for Research and Applications Version 2 (MERRA-2) data (e.g., Molod et al., 2015; Teng et al., 2021) to ensure consistency with observed meteorological conditions. Both chemical and dynamical parameters in SD-WACCM-X are relaxed toward linearly time-interpolated 3-hourly MERRA-2 reanalysis data. The relaxation coefficient is uniform below 50 km, decreases progressively above this altitude, and becomes zero above 60 km (Brakebusch et al., 2013). Consequently, the model is unconstrained by reanalysis above 60 km. The vertical resolution of the model ranges from $\sim 1.1$ to $\sim 3.5$ km, with a vertical density of two points per scale height below $\sim 50$ km and increasing to four points per scale height above $\sim 50$ km (Salinas et al., 2023). Enhanced vertical resolution in the troposphere and stratosphere enables improved representation of key physical processes (Sassi and Liu, 2014). The horizontal resolution is typically set to $1.9° \times 2.5°$ (latitude $\times$ longitude). It should be noted that CESM currently provides emission input files for the SD-WACCM-X only up to the year 2015 (https://svn-ccsm-inputdata.cgd.ucar.edu/trunk/inputdata/atm/cam/chem/emis/CMIP6_emissions_1750_2015_2deg, last access: 1 September 2024). Accordingly, prescribed emissions from CESM input were used prior to 2015, while for subsequent years, the emission fields were set to missing values. Sensitivity tests conducted for representative pre-2015 years (2009, 2010, and 2013) demonstrate that this treatment has a negligible impact on the simulated mesospheric OH response to SSW (Fig. S1 in the Supplement), thereby ensuring the robustness of our results.

SD-WACCM-X simulations have been demonstrated to effectively reproduce observed atmospheric responses to ES-SSW events (e.g., Limpasuvan et al., 2016; Lee et al., 2021; Zhang et al., 2021, 2025; Orsolini et al., 2022). These successes highlight the model's robustness in capturing dynamical and chemical processes, particularly within the MLT. In this study, model outputs spanning the period from 2004 to 2023 are utilized, with data from December through March extracted for each year, corresponding to the climatological window during which ES-SSW events predominantly occur. To comprehensively elucidate the response of the OH layer on ES-SSW, key variables such as OH, atomic oxygen, temperature, zonal wind, and so on are output. From these outputs, daily means of relevant dynamical and chemical parameters are derived to facilitate temporal-spatial evaluation of ES-SSW responses.

OH emission measurements from the SABER instrument onboard the TIMED satellite (https://saber.gats-inc.com/data.php, last access: 1 September 2024) are employed to validate, in a qualitative and time-evolution sense, the SD-WACCM-X OH response to ES-SSWs. TIMED satellite was launched on 7 December 2001 and began SABER observations in early 2002. SABER provides global limb observations of OH airglow in the mesosphere and lower thermosphere, with latitude coverage alternating every 60 d between 53 and 83° in the opposite hemisphere (Gao et al., 2011, 2016).

## 2.2 Definition of SSW and ES Events

According to the criteria established by McInturff (1978), a major SSW is identified when, at 10 hPa or below, the latitudinal mean temperature must increase poleward of 60° latitude, accompanied by a reversal in the zonal-mean zonal winds (i.e., a transition from mean westerly to mean easterly winds poleward of 60° latitude). However, the definition of major SSWs has evolved over the decades and varies across studies, ranging from Northern Annular Mode index values (Baldwin and Dunkerton, 2001) to classifications emphasizing vortex morphology (Charlton and Polvani, 2007), or disturbances centered near the stratopause ($\sim 50$ km) rather than the canonical 10 hPa level (Tweedy et al., 2013; Stray et al., 2015; Limpasuvan et al., 2016). Given these inconsistencies, we define the SSW onset as the day when the temperature difference between 60–70 and 80–90° N ($T[80–90°]$– $T[60–70°]$ K) at 10 hPa becomes positive, a diagnostic also adopted in Ma et al. (2020).

An elevated stratopause (ES) event is commonly described as a disruption in which the winter polar stratopause initially descends, then becomes indistinct, and eventually reforms at a much higher altitude than usual (Chandran et al., 2013). Despite broad recognition of the phenomenon, a community-wide, internally consistent definition has not yet been established. Consequently, the diagnostic criteria for identifying ES events remain ambiguous, with different studies applying different thresholds. For example, de la Torre et al. (2012) required the newly reformed stratopause to be displaced by more than 15 km relative to its original altitude, whereas Tweedy et al. (2013) and Harvey et al. (2025) used a threshold of a polar-cap stratopause elevation exceeding 10 km. Yamazaki et al. (2025) defined an elevated stratopause (ES) event as a stratopause detected above 80 km during December–February. In this study, any upward displacement of the stratopause relative to its pre-warming altitude is regarded as an ES event. Applying this criterion, ten ES-SSW events are identified over 2004–2023. Notably, the 2010 and 2018 SSWs with weak stratopause elevations have been treated as ES-SSWs in several studies (e.g., Harvey et al., 2025; Schneider et al., 2025; Harada et al., 2019). Guided by stratopause morphology similarity, the 2021 and 2023 warmings are also classified as ES-SSW in our com-

posite analysis, and both display weak stratopause elevations. The stratopause height is defined as the altitude of maximum temperature within the 20–100 km vertical domain (Chandran et al., 2013; de la Torre et al., 2012). Nearly all identi-
fied ES events in our study are associated with major SSWs, with the only exception being the 2012 case, which followed a minor warming but still exhibited a pronounced elevated stratopause.

To develop a statistically robust understanding of the OH
layer characteristics during ES-SSW, a composite analysis is conducted by temporally aligning each event such that Day 0 corresponds to the SSW onset. This alignment allows for a systematic examination of the temporal evolution and spatial structure of OH layer during ES-SSW events, providing
insights into their common features and variability. In this study, the response of OH layer to ES-SSW events is categorized into three distinct stages based on the temporal evolution of temperature: Day −15 to Day 0 is considered the normal stage; Day 0 to Day 5 correspond to the stratosphere
warming stage; Day 6 to Day 60 correspond to the elevated stratopause stage. Previous studies (e.g., Pickett et al., 2006) demonstrated that the OH layer is markedly more stable in pressure coordinates than in geometric altitude. Meanwhile, as noted by Grygalashvyly et al. (2014), pressure coordinates
are more natural for discussing chemical, physical, and radiative processes, as these processes are largely free from the shrinking of middle atmospheric (SMA) effects on pressure isosurfaces. Guided by these findings, we use log-pressure height ($Z = -H \times \ln(P/P_s)$) (Andrews et al., 1987) as the
vertical coordinate for diagnosing the OH responses in SD-WACCM-X, while also providing an approximate geometric-height scale for reference.

## 3  Results

The SSW event that commenced in January 2009 started as
the strongest, most prolonged on record, and isolated, which has been made a focal point for numerous studies (e.g., Manney et al., 2009; Yue et al., 2010; Limpasuvan et al., 2011). These unique characteristics render it particularly suitable for employing the response of mesospheric chemical com-
position to SSW events. The response of OH concentrations to ES-SSW events is exemplified by the 2009 SSW case, as illustrated in Fig. 1. From top to bottom, Fig. 1 represents (a) zonal-mean temperature at 10 hPa (∼ 32 km) and 80° N, (b) zonal-mean zonal wind at 60° N, (c) temporal
variations in OH concentrations at latitudes 75∼ 90° N from SD-WACCM-X, and (d) in OH emission at 75–77° N from SABER during 6 January to 22 March 2009. In Fig. 1a, the meridional temperature difference between 60–70 and 80–90° N ($T[80–90°]$–$T[60–70°]$ K) is denoted by the pink solid
line, and the stratopause height is indicated by the green dashed line.

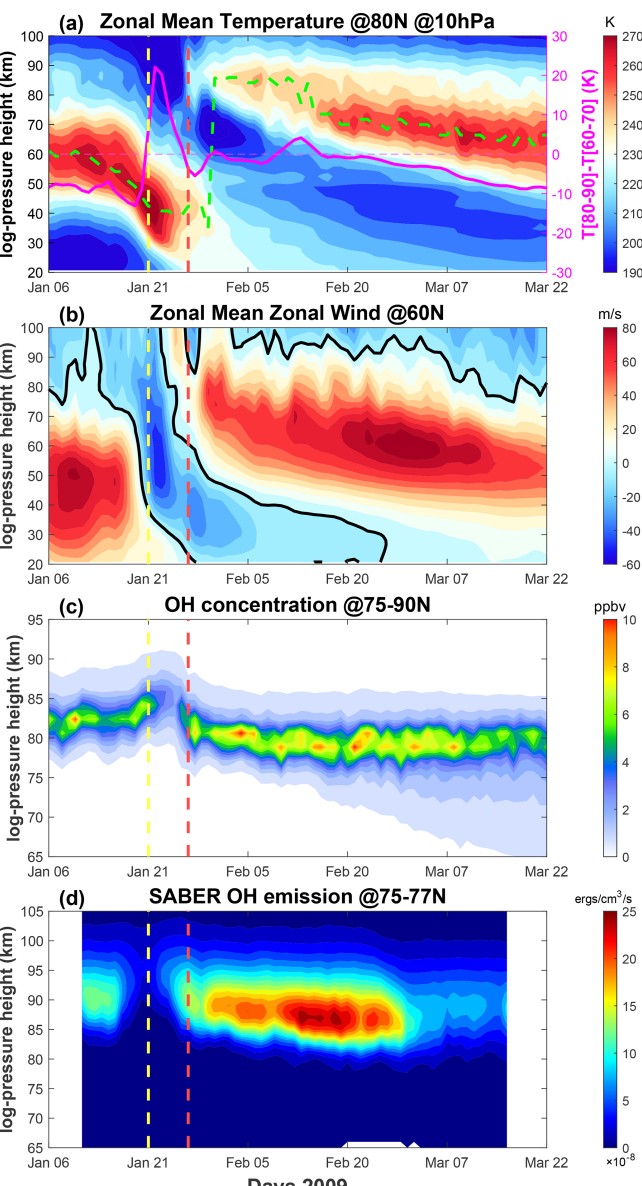

**Figure 1.** Time versus log-pressure height cross-section of SD-WACCM-X zonal-mean **(a)** temperature at 80° N, **(b)** zonal wind at 60° N, **(c)** OH concentration, and SABER **(d)** OH emission, during 6 January–22 March 2009. The solid pink line represents the meridional temperature difference between 60–70 and 80–90° N ($T[80–90°]$–$T[60–70°]$), and the dashed green line is the stratopause height. The black solid contour in panel **(b)** denotes the zero-wind line. Vertical dashed yellow and red lines are the onset of the stratosphere warming stage and elevated stratopause stage, respectively.

Prior to 10 January 2009, the temperature peak is located at ∼ 61 km, with a maximum ∼ 272 K. Subsequently, the peak height drops sharply, reaching ∼ 37.7 km by 21 January 2009. Between 21 and 26 January 2009, the minimum
temperature of ∼ 183.5 K occurs at altitudes above 75 km, while the meridional temperature difference between 60–70

and 80–90° N at 10 hPa transitions to positive values. After 26 January 2009, a new stratopause forms at $\sim 86$ km, with an upward displacement of $\sim 25$ km compared to the stratopause altitude before 10 January 2009, and subsequently descends to its climatological altitude in conjunction with the recovery of the polar vortex (Limpasuvan et al., 2012). This elevated stratopause event is accompanied by a pronounced downwelling over the winter pole, extending from 45 to 95 km. The enhanced subsidence facilitates the downward transport of MLT air into the stratosphere, thereby modulating the thermal structure and influencing polar chemical processes.

Figure 1b depicts the zonal mean zonal wind changes at 60° N. Before the SSW onset, the zonal-mean zonal wind is eastward with a speed of $\sim 85$ m s$^{-1}$. During the stratosphere warming phase, the zonal mean flow reverses to westward winds (below 0 m s$^{-1}$) in the polar winter stratosphere, confirming the occurrence of a major SSW event. Subsequently, the zonal-mean zonal wind structure during the elevated stratopause exhibits a pattern similar to that of the zonal-mean temperature, with eastward winds prevailing in the mesosphere region at $\sim 91.8$ m s$^{-1}$. As the polar vortex recovers, the eastward wind gradually descends to its climatological distribution, mirroring the temperature evolution in the stratosphere and mesosphere.

Figure 1c illustrates the temporal variation of OH concentration during the 2009 events. Prior to the onset, the OH concentrations remain $\sim 7.2$ ppbv with a peak height $\sim 82.4$ km, consistent with previous studies (e.g., Pickett et al., 2006). However, between 21 and 26 January 2009, a pronounced depletion in OH concentrations is observed reaching a minimum of $\sim 2.4$ ppbv, while the peak altitude shifts upward to $\sim 86$ km. Following 26 January 2009, the peak altitude of OH layer descends to $\sim 78.8$ km, whereas in geometric coordinates the descent is even larger, reaching $\sim 74$ km (not shown). Meanwhile, the peak OH concentrations exhibits a gradual increase, eventually surpassing the pre-SSW levels and reaching values of $\sim 10.6$ ppbv. The anomaly persisted and did not return to its climatological level until 9 March 2009.

The OH emission retrieved by SABER (Fig. 1d) exhibits a temporal evolution broadly consistent with the SD-WACCM-X OH concentration. This correspondence is expected because their shared production mechanism via the $O_3 + H \xrightarrow{k_1}$ OH* + O$_2$, which forms vibrationally excited OH $\sim 90$ km (e.g., Grygalashvyly et al., 2014) and collisional quenching at lower altitudes produces ground-state OH near 82 km (e.g., Damiani et al., 2010). Although the two quantities are not physically identical, they are dynamically linked through the same mesospheric circulation and chemistry. For example, Gao et al. (2011) and Winick et al. (2009) demonstrated that variation in OH emission during the 2009 SSW was modulated by changes in atomic oxygen and mesospheric temperature, and Damiani et al. (2010) showed that the displace-

ment of OH layer was coupled to variations in odd-oxygen (O$_x$) and mesospheric temperature. Accordingly, we restrict the comparison to qualitative temporal evolution rather than absolute magnitudes to evaluate general consistency between model simulations and satellite observations during ES-SSW events.

Notably, to ensure consistency with the SD-WACCM-X simulations, the SABER OH emissions are also mapped onto log-pressure height, which yields a peak altitude that is slightly higher ($\sim 90$ km) than those reported in earlier studies ($\sim 87$ km) based on geometric height. This offset is consistent with Grygalashvyly et al. (2014), who compared the OH layer in geometric and log-pressure coordinates and showed that the apparent differences arise because geometric altitudes are affected by SMA, whereas pressure coordinates follow approximately fixed mass surfaces and are largely free from this effect. As a result, the OH peak expressed in log-pressure height tends to appear somewhat higher than when it is expressed in geometric height.

To further validate the observed response of OH concentrations to ES-SSW events, SSW occurrences from 2004 to 2023 are analyzed, aiming to identify common characteristics across multiple events. Based on the criteria outlined in the methodology section, a total of ten ES-SSW events are identified (Manney et al., 2008; Maute et al., 2014; Kodera et al., 2008; Manney et al., 2009; Harada et al., 2010; Jones et al., 2018; Chandran et al., 2013; Goncharenko et al., 2013; de Wit et al., 2014; Karpechko et al., 2018; Rao et al., 2019; Okui et al., 2021; Lu et al., 2021; Qin et al., 2024; Zhang et al., 2025), with their occurrence years and onset dates listed in Table 1. The left column of the table indicates the years in which ES-SSW events occurred, while the right column specifies the first day of each event.

Figure 2 presents the temporal variations in OH concentrations for each selected year, centered on the onset day of the ES-SSW event and spanning from 15 d prior to 60 d after the onset. As shown, during the stratosphere warming phase, the peak height of OH concentration undergoes an evident upward displacement to $\sim 86$ km, whereas the OH concentrations peak experiences a sharp decline with the minimum of $\sim 1$–2 ppbv. Conversely, during the elevated stratopause phase, as the peak height of the OH layer significantly decreases, the peak OH concentration increases rapidly to $\sim 7$–9 ppbv at $\sim 78$ km, and its maximum is larger than that in the normal stage. The results reveal a consistent pattern similar to that shown in Fig. 1c, demonstrating that the influence of ES-SSW events on OH concentration is a common feature across all identified occurrences. In certain years, i.e., 2010, 2018, and 2023, the influence of ES-SSW events on the structure of OH concentration appears weaker, potentially due to variations in event intensity or background atmospheric conditions. Furthermore, although SSW events vary in their duration and intensity, they tend to occur on an annual basis. Some events occur in close succession, as observed in 2008, whereas others appear as isolated episodes, such as the one

**Table 1.** TS2 Onset Day (Day 1 corresponds to 1 January of the Year) of the ES-SSW Events During Boreal Winter of 2004–2023.

| Years | Onset day | Years | Onset day |
|-------|-----------|-------|-----------|
| 2003/2004 | 2 January | 2012/2013 | 5 January |
| 2005/2006 | 9 January | 2017/2018 | 14 February |
| 2008/2009 | 21 January | 2018/2019 | 25 December |
| 2009/2010 | 20 January | 2020/2021 | 2 January |
| 2011/2012 | 11 January | 2022/2023 | 13 February |

Note. ES-SSW, sudden stratosphere warming with elevated stratopause.

in 2009. During ES-SSW events, preceding or subsequent SSW-related mesospheric warming and elevated stratopause phases remain active, introducing additional modulation to the observed OH variations.

Notably, OH concentrations in some SSW events such as February 2010 and 2023, also show a significant downward extension in altitude after March, suggesting that seasonal variability may play a noticeable role in the temporal evolution of OH concentrations. As illustrated in Fig. 3, OH concentrations display a clear seasonal pattern, with higher values in the summer hemisphere (May to August in the Northern Hemisphere; November to February in the Southern Hemisphere) and lower values in the winter hemisphere. The peak of OH concentration in the summer hemisphere reaches $\sim 12.5$ ppbv, whereas the minimum OH concentration in the winter hemisphere is $\sim 5$ ppbv. Additionally, the seasonal cycle also modulates the vertical structure of the OH layer: in summer, OH extends broadly from $\sim 75$ to $\sim 87$ km, whereas in winter it is confined near 82.5 km. This summer broadening helps explain the downward extension of OH observed during late-winter sudden stratospheric warming events, such as those in 2010 and 2023.

Figure 4 illustrates the composite evolution of zonal mean temperature, zonal mean zonal wind, OH concentration, and OH concentration anomaly (Anomaly derived from original OH concentration minus background average) as functions of altitude and time. The background average of OH concentration is derived from the 20-year average from 2004 to 2023. As depicted in Fig. 4a and b, significant variations extend across nearly the entire altitude range. In the mesosphere, the zonal mean zonal wind exhibits a rapid weakening of the eastward component around Day $-8$, then switches to westward after Day 0 with a minimum wind speed of around $-27$ m s$^{-1}$. Around Day 6, the eastward wind begins to intensify again, reaching a peak velocity exceeding 70 m s$^{-1}$ near Day 40. Corresponding thermal changes accompany zonal wind reversal. At the onset of the ES-SSW event, the stratosphere undergoes rapid warming, leading to a sharp descent of the stratopause altitudes. As shown, until Day 3, the stratopause altitude reaches its lowest altitude of $\sim 43.5$ km, coinciding with the peak westward wind. The maximum temperature and wind speed are over 260 K and $-27$ m s$^{-1}$, respectively. After Day 6, as the eastward wind

begins to strengthen, a newly formed stratopause emerges near 80 km.

The composite variabilities of OH concentrations and OH concentrations anomaly associated with ES-SSW events are examined, as shown in Fig. 4c and d. Here, the composite results represent the mean structure aligning the temporal series of individual ES-SSW events. Figure 4c illustrates the temporal evolution of OH concentration. Prior to the onset of SSW, the peak of OH concentration exhibits a value of $\sim 7.4$ ppbv, with a peak height near 82.4 km. During the stratosphere warming phase, the peak of OH concentration decreases to 2.9 ppbv, while in the elevated stratopause phase, it increases to $\sim 6.8$ ppbv. In addition, the peak height of OH concentration rises by $\sim 3.5$ km reaching $\sim 85.9$ km during the warming phase, before descending by $\sim 2$ to $\sim 80.6$ km during the elevated stratopause phase. This temporal evolution is consistent with that in 2009 shown in Fig. 1. Winick et al. (2009) proposed that the anomalous characteristics of the OH layer may be associated with changes in the atomic oxygen concentration and temperature in the mesosphere, which is driven by the modification of polar circulation induced by SSW events.

Similarly, the temporal evolutions are depicted in Fig. 4d, which represents the variability of OH concentration anomaly. As shown, during the stratosphere warming phase, the peak height of OH concentration anomaly occurs at $\sim 85.9$ km with a maximum of $\sim 1.3$ ppbv. These situations in the elevated stratopause phase are significantly different. Compared to the stratosphere warming phase, the peak of OH concentration anomaly doubles, reaching a maximum of $\sim 2.6$ ppbv, while its peak height significantly decreases to $\sim 78$ km. These phenomena again demonstrate the consistency between OH concentration enhancements/depletions and peak height descent/ascent, a relationship previously documented in Winick et al. (2009).

## 4   Discussion

### 4.1   Temporal variation

During the ES-SSW period, changes in atmospheric circulation lead to a strong descent/ascent motion of air at high latitudes. This dynamic change leads to subse-

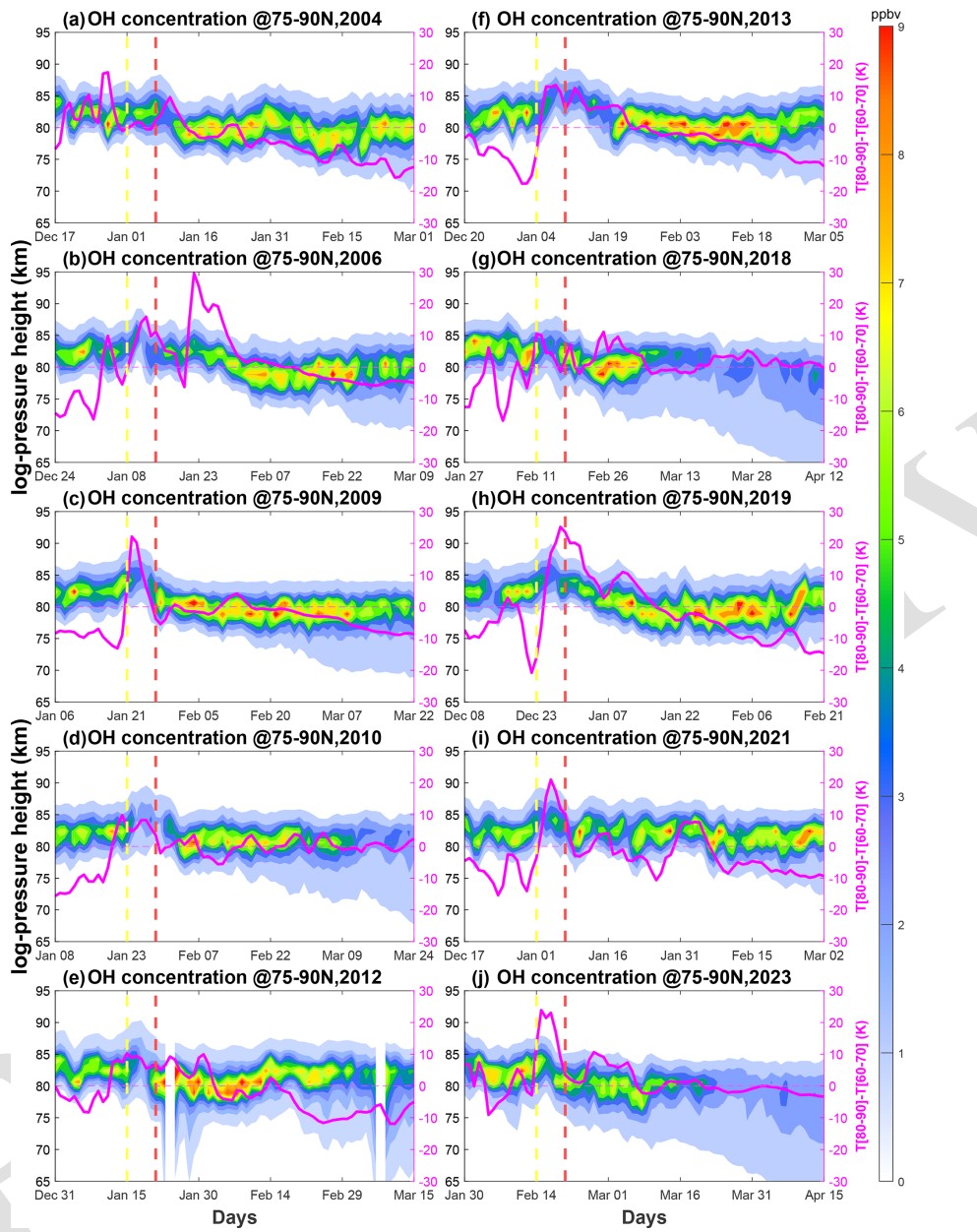

**Figure 2.** Time versus log-pressure height cross-sections of OH concentrations in ES-SSW events during 2004–2023 are captured in panels **(a)**–**(j)**. The solid pink line represents the meridional temperature difference between 60–70 and 80–90° N ($T[80$–$90°]$–$T[60$–$70°]$). Vertical dashed yellow and red lines are the onset of the stratosphere warming stage and elevated stratopause stage, respectively.

quent increased/decreased atomic oxygen penetrating the mesopause. Within the mesospheric region, OH* is primarily produced through the reaction between ozone ($O_3$) and atomic hydrogen (H) (Marsh et al., 2006), as described by this equation:

$$O_3 + H \xrightarrow{k_1} OH^* + O_2. \tag{1}$$

According to Eq. (1), the reaction rate coefficient $k_1$ increases with temperature, thereby enhancing the efficiency of OH* production. Ozone is formed through the reaction of atomic

oxygen and molecular oxygen, as expressed in the equation:

$$O + O_2 + M \rightarrow O_3 + M. \tag{2}$$

Notably, this process, $O + O_3 \rightarrow O_2 + O_2$, also contributes to ozone destruction alongside OH* production. However, below $\sim 95$ km, ozone loss due to its reaction with atomic hydrogen significantly exceeds that caused by atomic oxygen by several orders of magnitude (Xu et al., 2010). In addition, ozone may be treated as being in a steady state under nighttime conditions (i.e., polar winter), implying that the

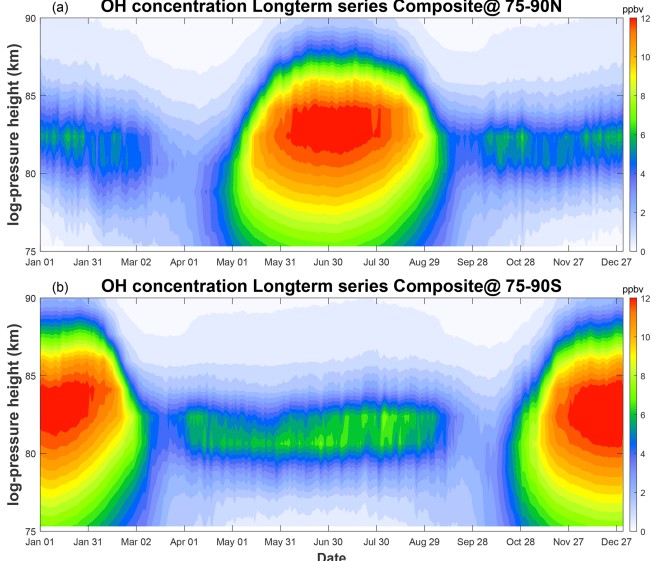

**Figure 3.** Temporal evolution of OH concentration in the polar region, with the Northern Hemisphere in the top panel and the Southern Hemisphere in the bottom panel.

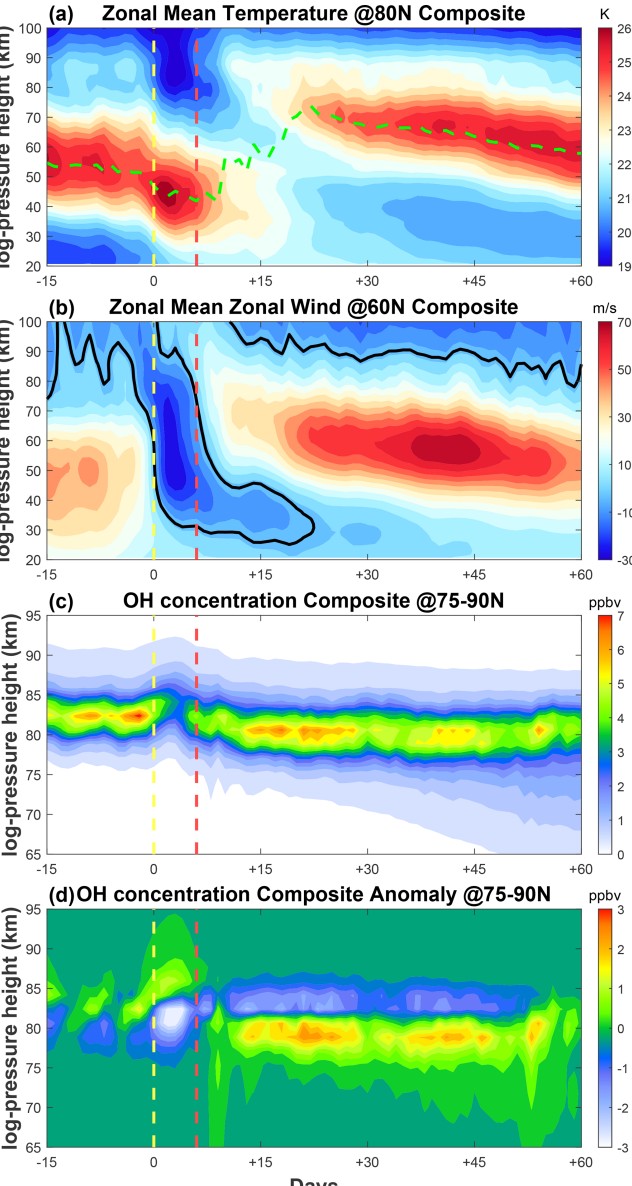

**Figure 4.** Time versus log-pressure height cross-section of composite zonal mean **(a)** temperature, **(b)** zonal wind, **(c)** OH concentration, and **(d)** OH concentration anomaly. On the abscissa, time is relative to the ES-SSW onset (Day 0). Vertical dashed yellow and red lines are the onset of the stratosphere warming stage and elevated stratopause stage, respectively. The dashed green line is the stratopause height, while the black solid contour in panel **(b)** denotes the zero-wind line.

production of OH is proportional to atomic oxygen (Gao et al., 2011). This relationship indicates that the temporal evolution of OH concentration is strongly coupled with variations in atomic oxygen abundance and temperature in the

5 mesopause.

Figure 5 presents the composite variations in atomic oxygen (Fig. 5a) and temperature (Fig. 5b) over high latitudes (75–90° N). In Fig. 5a, the solid black line illustrates the temporal evolution of atomic oxygen concentration at 82 km.

The atomic oxygen concentration increases with altitude in the mesopause. Since the atomic oxygen concentration is relatively low at this level, the line plot is included to clearly highlight its variation throughout the event. The temporal variations of OH (Fig. 4c), atomic oxygen (Fig. 5a), and tem-

perature (Fig. 5b) in the mesopause region exhibit a nearly synchronous evolution. During the stratosphere warming phase, the atomic oxygen concentration experiences a significant decline, with a minimum value of $\sim 5$ ppmv at $\sim 82$ km, as indicated by the solid black line. Then, the atomic oxygen

peak concentration remarkably increases, reaching a peak value of $\sim 240$ ppmv, which is substantially higher than the normal stage. Figure 5b displays the composite temperature variation, which closely corresponds to the evolution of atomic oxygen. The temperature minimum reaches its min-

imum ($\sim 190$ K) during the stratosphere warming stage and peaks at $\sim 260$ K in the elevated stratopause stage.

Figure 6 shows the temporal evolution of the residual circulation ($v^* = \overline{v} - \rho^{-1}\left(\rho \frac{\overline{v'\theta'}}{\overline{\theta}_z}\right)_z$; $w^* = \overline{w} + (a\cos\varphi)^{-1}\left(\cos\varphi \frac{\overline{v'\theta'}}{\overline{\theta}_z}\right)_{\varphi'}$) (Andrews et al., 1987),

where $w^*$ and $v^*$ denote the vertical and meridional com-

ponents of the residual circulation, respectively. During the stratosphere warming phase, a strong downwelling (negative $w^*$) develops in the stratosphere with a value of $\sim -1$ cm s$^{-1}$, denoting enhanced adiabatic heating. Meanwhile, an anomalous upwelling (positive $w^*$) emerges

above $\sim 70$ km in the mesosphere, peaking at $\sim 1.4$ cm s$^{-1}$ at $\sim 77$ km, suggesting a weakening or even reversal of the

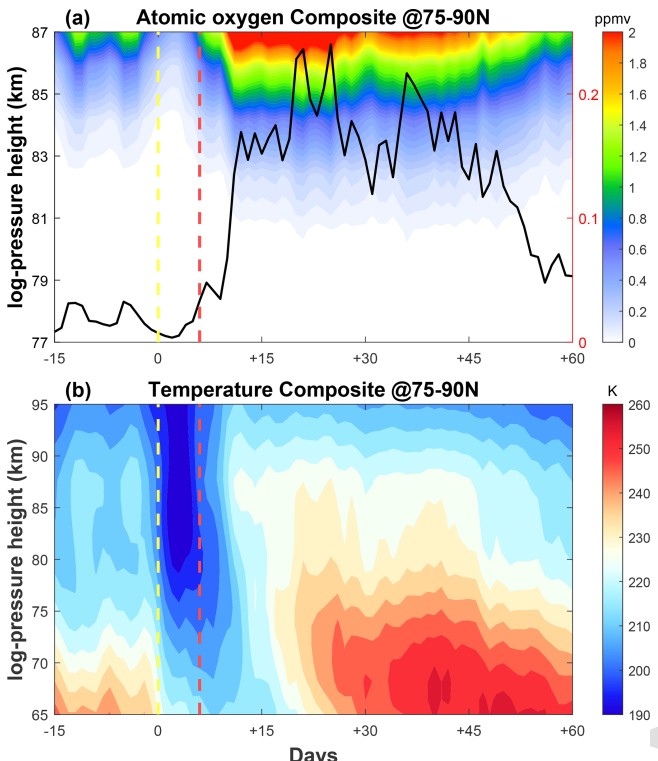

**Figure 5.** Same as Fig. 4, but for **(a)** atomic oxygen (in units of $10^3$) and **(b)** temperature. The solid black line in panel **(a)** represents the temporal evolution of atomic oxygen at 82 km.

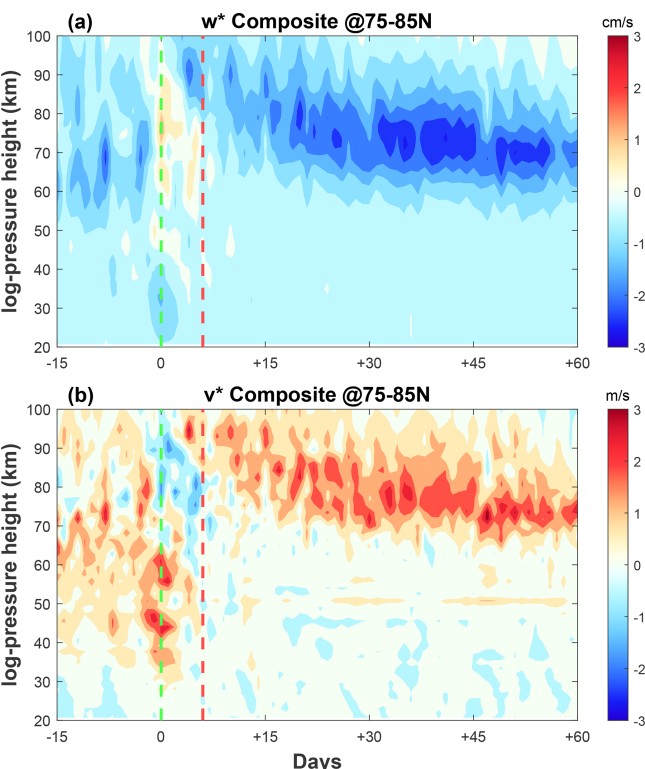

**Figure 6.** Same as Fig. 4, but for **(a)** $w^*$ and **(b)** $v^*$. Vertical dashed green and red lines are the onset of the stratosphere warming stage and elevated stratopause stage, respectively.

climatological downward branch of the residual circulation over the winter polar cap (e.g., Gao et al., 2011; Limpasuvan et al., 2016). These vertical circulation anomalies directly modulate the mesospheric thermal structure: the upwelling leads to adiabatic cooling, consistent with the temperature decrease observed in Fig. 4a, while the subsequent recovery of strong downwelling in the elevated stratopause phase (up to $-2.5\,\mathrm{cm\,s^{-1}}$) contributes to mesospheric warming. Additionally, the upwelling during the stratosphere warming phase lifts air with lower atomic oxygen concentrations into the mesopause, resulting in a significant depletion of atomic oxygen (Fig. 5a). In contrast, enhanced downwelling during the elevated stratosphere phase brings oxygen-rich air downward, increasing atomic oxygen concentration and thereby promoting OH production. During the stratosphere warming stage, an anomalous equatorward flow also emerges near the mesopause (Fig. 6b), indicating a temporal reversal of the climatological poleward residual circulation. As the stratopause elevates, this meridional circulation returns to a poleward pattern. The reversal and recovery of meridional circulation could promote inter-latitudinal transport, potentially contributing to OH variations at lower latitudes, such as the equatorial region.

## 4.2 Spatial distribution

As shown in the preceding figures, the most prominent variations in OH concentration associated with ES-SSW event occur primarily within the first 30 d following the SSW onset, during which OH, atomic oxygen, temperature, and $w^*$ exhibit substantial perturbations. To further investigate the meridional structure of these responses, the relative variations of these parameters are analyzed across three distinct stages: Stage 1 (Day $-10$ to Day 0), Stage 2 (Day 0 to Day 5), and Stage 3 (Day 6 to Day 26). Figures 7–9 present the corresponding spatial distributions of each parameter during these stages. The relative variation is derived as follows:

$$\text{Reletive variation} = \frac{X_{(i)} - \overline{X}}{\overline{X}} \times 100\,\% \qquad (3)$$

in which $X_{(i)}$ represents the variable (OH, atomic oxygen, and temperature) for each ES-SSW event, and $\overline{X}$ stands for the background average obtained by averaging the same calendar dates over 2004–2023.

Figure 7 illustrates the ES-SSW-related parameters as functions of latitude and altitude in Stage 1. The spatial structure of OH concentration in the polar mesosphere is depicted in Figure 7a. A local maximum appears near 83 km with an enhancement of 10 % relative to the background average, whereas below 80 km, OH concentration exhibits

a marked decrease, reaching a minimum of $-35.7\%$. Figure 7b presents the situation of the atomic oxygen, which displays notable differences from OH. For instance, the enhancement shifts downward, with a peak of $25.4\%$ located at $\sim 75\,\mathrm{km}$, while the minimum atomic oxygen concentration reaches $-46.3\%$ at $\sim 81\,\mathrm{km}$ in the polar region. The corresponding temperature distribution (Fig. 7c) reveals a warming in the polar stratosphere and cooling in the polar mesosphere with the magnitudes of approximately $3\%$ and $-3\%$, respectively. Figure 7d shows the $w^*$ anomaly, which increases slightly to $\sim 0.4\,\mathrm{cm\,s^{-1}}$ between $\sim 65$ and $83\,\mathrm{km}$ at high latitude. Simultaneously, the polar $v^*$ anomaly, as indicated by the white arrows, shifts equatorward in the mesosphere. In climatology, the circulation pattern is characterized by a pole-to-pole circulation from the summer hemisphere toward the winter hemisphere. Notably, slight variations in these parameters can be detected even prior to the onset of the ES-SSW event. This early-stage response may be attributed to the fact that ES-SSW perturbations tend to emerge in the mesosphere several days before becoming evident in the stratosphere (Gao et al., 2011), as also reflected by the temperature variations in Fig. 4a.

Figure 8 illustrates the latitude-altitude structure of various elements in Stage 2, which exhibit significant responses to ES-SSW events. Compared to Stage 1, the OH concentration (Fig. 8a) in the polar region displays a marked enhancement with a maximum increase of $\sim 75.9\%$ relative to the background average occurring at $\sim 85.9\,\mathrm{km}$. Conversely, the minimum OH concentration reaches a pronounced decrease with an amplitude of $-63.5\%$ at $\sim 78.8\,\mathrm{km}$. The variations in atomic oxygen (in Fig. 8b) are even more pronounced concerning Stage 2 over the latitudinal range of 75–90° N, with a minimum reduction to $-90.8\%$ and a maximum increase to $\sim 48.3\%$ with respect to the background average. The temperature distribution in Fig. 8c also reveals a noticeable response to the ES-SSW events, with a positive anomaly of $13.9\%$ centered near $32\,\mathrm{km}$ and a negative anomaly of $-11\%$ around $84\,\mathrm{km}$ in the polar region. Figure 8d shows the spatial structure of $w^*$ anomaly in Stage 2. Compared to Stage 1, the magnitude of $w^*$ anomaly in the polar mesosphere increases significantly, exceeding $1.78\,\mathrm{cm\,s^{-1}}$ between $\sim 64.6$ and $\sim 78.8\,\mathrm{km}$. These vertical structures are consistent with the findings of Dyrland et al. (2010) that associated OH airglow temperature perturbations with neutral atmospheric dynamics during the anomalous 2003–2004 winter. Their study also noted altitude-dependent evolution of the polar vortex.

In the present work, the positive $w^*$ anomaly at high northern latitudes is clearly observed, as indicated by the white arrows in Fig. 8d. This upward motion turns equatorward in the upper mesosphere, forming a distinct pole-to-equator branch of the mesospheric circulation. Simultaneously, below $\sim 50\,\mathrm{km}$, the poleward circulation exhibits a downward motion at the middle and high latitudes. The high-latitude downward circulation below $\sim 50\,\mathrm{km}$ and upward circulation above $\sim 50\,\mathrm{km}$ result in corresponding stratospheric warming and mesospheric cooling during the SSW period (as also shown in Fig. 4a), respectively. Moreover, the upward transport of oxygen-poor air to higher latitudes, facilitated by positive $w^*$ anomaly, results in a significant reduction in atomic oxygen concentrations. As a consequence, the peak height of OH layer shifts upward, accompanied by a notable decrease in its peak concentration.

In addition, notable variations in OH concentration are also evident in the equatorial mesosphere region during Stage 2. Relative to the background average, a secondary OH peak appears at $\sim 78.8\,\mathrm{km}$ with an amplitude of $\sim 13.6\%$, while a local minimum of approximately $-16.9\%$ occurs at a higher altitude of $\sim 82.4\,\mathrm{km}$. Concurrently, both atomic oxygen and temperature exhibit modest enhancements in the equatorial mesosphere, with amplitudes of $\sim 16\%$ and $\sim 1.5\%$, respectively. The observed temperature response in the equatorial region is consistent with that reported by Gu et al. (2021) in their analysis of the 2009 major SSW event (see their Fig. 2e), which documented the middle atmospheric circulation response to SSW. However, the underlying dynamical and chemical mechanisms responsible for these compositional changes in the equatorial region remain unclear and warrant further investigation.

Figure 9 depicts the spatial distribution of multiple parameters during Stage 3, corresponding to Day 6 through Day 26 following the onset of the SSW. Compared to the preceding stages, the polar OH concentration structure undergoes a distinct reversal in vertical pattern. Specifically, the OH peak descends to $\sim 78.8\,\mathrm{km}$ and intensifies to $\sim 59.5\%$, while the minimum shifts to a higher altitude ($\sim 84.2\,\mathrm{km}$) and weakens to negative $37.7\%$. As shown in Fig. 9b, the atomic oxygen concentration exhibits a pronounced enhancement, with its peak occurring at $\sim 82.4\,\mathrm{km}$ and exceeding $137.3\%$ compared to the climatological average at high latitudes. The temperature field (Fig. 9c) displays a bifurcated structure, with a dominant warming centered around $\sim 28\,\mathrm{km}$ with an amplitude of $\sim 7.9\%$ and a secondary weaker warming near the MLT region with a lower amplitude of $\sim 3.7\%$. The vertical structure of the residual circulation anomaly (Fig. 9d) also changes noticeably. The peak of $w^*$ anomaly shifts downward to $\sim 57\,\mathrm{km}$ with a reduced amplitude ($\sim 0.51\,\mathrm{cm\,s^{-1}}$) in the polar region compared to earlier stages. As the mesospheric wind begins to eastward, the residual circulation anomaly transitions to a pole-to-pole pattern from the summer to winter hemisphere, and then descends vertically above $\sim 70\,\mathrm{km}$ in the polar region. The negative $w^*$ facilitates the downward transport of oxygen-rich air into the polar mesopause region and warming around $80\,\mathrm{km}$. Consequently, the OH concentration peak increases in magnitude while its altitude decreases. In the equatorial region, the OH distribution in Stage 3 resembles that in Stage 2 but with reduced amplitude. The sustained presence of these anomalies may be attributed to continued enhancements in

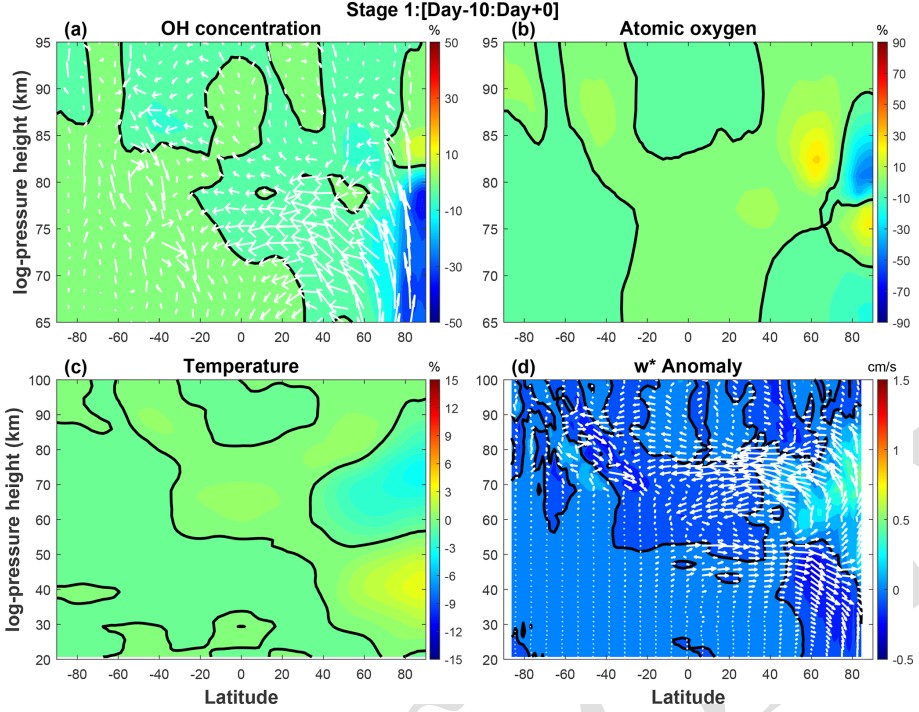

**Figure 7.** Latitude versus log-pressure height cross-sections of composite relative variation in (**a**) OH, (**b**) atomic oxygen, and (**c**) temperature, and composite anomaly in panel (**d**) $w^*$ during Day $-10$ to Day 0 (Stage 1). The zero contour is denoted by a bold solid black line, and white arrows in panels (**a**) and (**d**) represent the residual circulation anomaly.

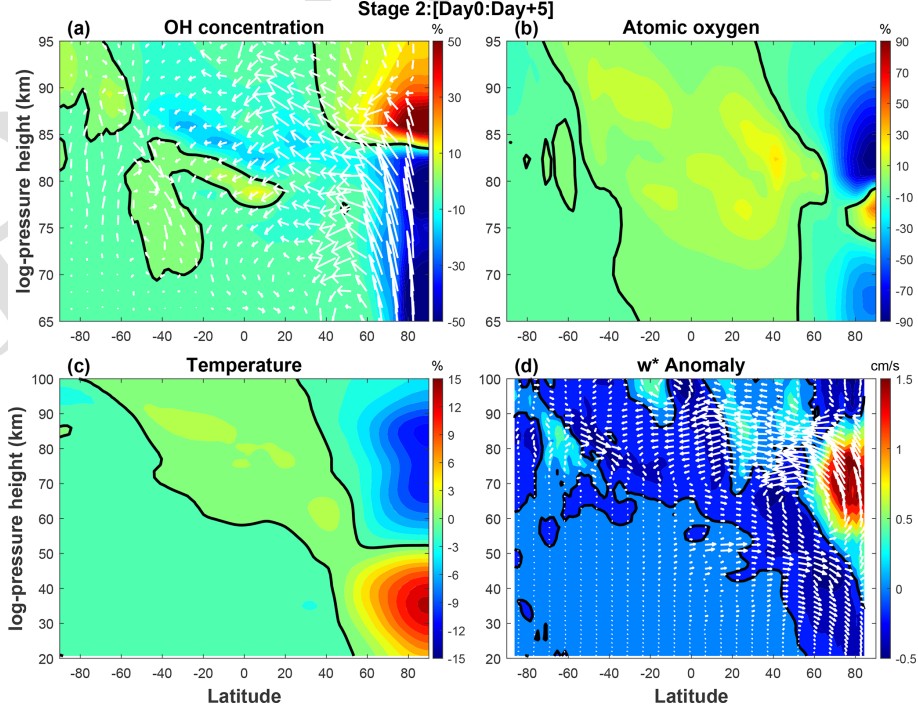

**Figure 8.** Same as Fig. 7, but for Day 0 to Day 5 (Stage 2).

both atomic oxygen abundance and temperature in the equatorial region.

## 4.3  Potential mechanism

During ES-SSW events, the reversal/deceleration of the stratospheric winter eastward wind due to planetary wave breaking allows more eastward-propagating GWs to penetrate into higher altitudes, enhancing upward and equatorward motion in the mesosphere. This altered circulation results in mesospheric cooling (Limpasuvan et al., 2012; Lukianova et al., 2015; Gu et al., 2021). The mesospheric OH layer variability may be dominantly controlled by changes in GWs during the ES-SSW events.

Figure 10 presents the temporal evolution of zonal wind tendency due to total gravity wave drag (GWD) and zonal wind tendency due to orographic gravity wave drag over the latitudes of 40–80° N. Under the quiet conditions, westward GWs drag in the mesosphere sustain the polar circulation downwelling. As planetary wave activity intensifies, the weakening/reversal of the eastward jet starts in the mesosphere and progresses to the stratosphere (Yang et al., 2024; Liu and Roble, 2005), with mesospheric changes preceding the stratospheric response by several days in Fig. 4b (Coy et al., 2011; Kurihara et al., 2010; Azeem et al., 2005). Alteration of the wave transmission conditions enables eastward-propagating GWs to penetrate the mesosphere resulting in enhanced eastward GWD that modulates the mesospheric circulation (Fig. 6) and induces a slight response in the polar OH layer (Fig. 7a).

Following the stratosphere warming phase, a reversal of the zonal-mean wind from eastward to westward in the polar stratosphere and mesosphere is evident in Fig. 4b. The stratospheric westward wind filters out westward phase speed components of GWs and facilitates the upward propagation of eastward-propagating GWs into the mesosphere, which results in anomalously eastward GWs exerting strong positive zonal wind tendency as shown in Fig. 10a, peaking in the mesosphere around Day 4 with its maximum of $46.2\,\mathrm{m\,s^{-1}\,d^{-1}}$. Enhanced eastward GWs forcing induces equatorward flow in the mesopause region and enhances the upward residual circulation extending from the stratosphere to the mesosphere at northern high latitudes, which can be seen vertical component and meridional component of residual circulation in Figs. 6 and 8d. This upward residual circulation transports oxygen-poor air upward, given that atomic oxygen concentration increases with altitude (seen in Fig. 5a), leading to a reduction in atomic oxygen abundance in the upper atmosphere. As a result, the peak OH concentration exhibits a substantial decrease, while its corresponding peak altitude undergoes a pronounced upward shift.

During the elevated stratopause phase, the thermal relaxation rate in the upper mesosphere and lower stratosphere is approximately two to three times that of the lower stratosphere (Wehrbein and Leovy, 1982; Chandran et al., 2014),

promoting the rapid recovery of the polar eastward zonal wind (Hitchman et al., 1989). Consistent with the initial response to the disturbance, the recovery of eastward zonal winds first appears in the mesosphere and then progressively propagates downward into the stratosphere. The GWs with westward phase speeds are again able to propagate into the mesosphere and subsequently dissipate, inducing a pronounced negative zonal wind tendency with a peak value reaching $\sim -45\,\mathrm{m\,s^{-1}\,d^{-1}}$ exceeding the climatological average. Enhanced GW breaking at higher latitudes within the MLT region drives an intensified poleward and downward branch of the residual circulation, leading to the reformation of elevated stratopause at high latitudes, as shown in Fig. 4a (Chandran et al., 2014). Compared to quiet conditions, where such circulation is largely confined below 50 km, the poleward and downward flow extends above 70 km in the elevated stratopause phase. This pronounced downward motion promotes the descent of dry mesospheric air (Orsolini et al., 2010), which causes an exceptionally strong vertical transport of atomic oxygen into the mesopause region. Consequently, the OH concentration returns toward the climatological level, with the peak of OH layer occurring at a lower altitude.

Generally, GWs can be categorized into orographic and non-orographic components (Gilli et al., 2020; Richter et al., 2010). As shown in Fig. 10b, the momentum deposition induced by orographic GWs is primarily confined to the altitude range of 40–80 km with values ranging from $-10.5$ to $0.5\,\mathrm{m\,s^{-1}\,d^{-1}}$. During the stratosphere warming phase, orographic GWs are largely filtered and inhibited from upward propagation, leading to a notable reduction of their drag contribution (Liu et al., 2019). In the elevated stratopause phase, the zonal wind tendency associated with orographic GWD shows a different vertical structure compared to that associated with total GWD (see Fig. 10a), particularly exhibiting a less significant enhancement at higher altitudes. It is demonstrated that orographically generated GWs play a secondary role in the overall momentum budget relative to non-orographic GWs. Non-orographic GWs, which are parameterized mainly from frontogenesis and convection sources (Holt et al., 2017), dominate the momentum deposition in the mesospheric polar region during ES-SSW events. The contribution from convective GWs is comparatively negligible, while frontogenetical generated GWs emerge as the primary source of GWs drag (Limpasuvan et al., 2012, 2016). Given the dominant role of frontogenesis GWs in driving mesospheric dynamics during ES-SSW events, it is suggested that the variability observed in the OH layer is predominantly modulated by gravity wave activity associated with frontogenesis.

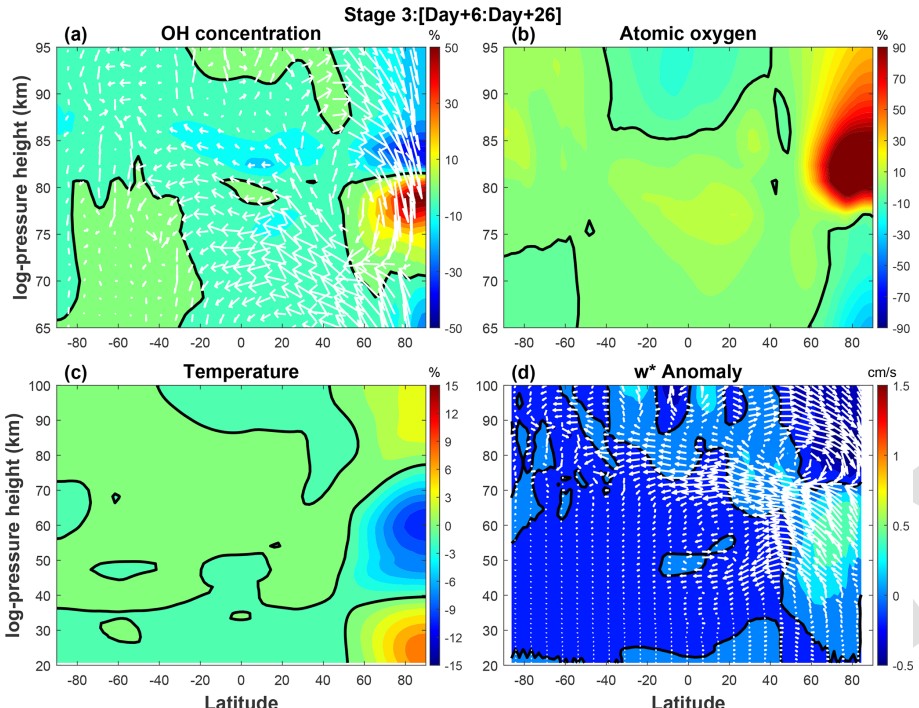

**Figure 9.** Same as Fig. 7, but for Day 6 to Day 26 (Stage 3).

## 5  Conclusions

In this paper, the responses of the peak concentration and peak height changes in the polar OH layer to ES-SSW events in the mesosphere are investigated based on SD-WACCM-X simulations. By compositing ten ES-SSW events from 2004 to 2023, distinct variations in OH layer structure associated with different phases of these events are revealed. The results demonstrate that the anomalous behaviors of the OH concentrations are closely synchronous with changes in mesospheric temperature, atomic oxygen concentrations, and the vertical component of the residual circulation in the MLT region. GWs play a pivotal role by altering the vertical motion of circulation in the MLT region, which modulates the zonal wind and temperature fields. The enhanced downward/upward motion driven by GWs leads to mesospheric warming/cooling and an increase/decrease in atomic oxygen, which facilitates an increase/decrease in OH concentration.

The impact of ES-SSW extends well from the polar to the equator, strongly altering the zonal-mean zonal wind, temperature, and atomic oxygen distribution. However, the response of the mesopause OH layer in the equatorial region to ES-SSW is unclear. Our study enhances the understanding of OH layer responses to stratospheric perturbations and provides new insights into vertical coupling processes in the middle and upper atmosphere. Future research should focus on quantifying the relative contributions of these factors and assessing their implications for long-term atmospheric dynamics and chemistry.

**Data availability.** WACCM-X is an open-source software with source code publicly available at https://escomp.github.io/CESM/release-cesm2/downloading_cesm.html#downloading-the-code-and-scripts (last access: 1 September 2024). The atmospheric forcing data, which are regridded from the MERRA-2 data set and used to run WACCM-X, can be downloaded at https://rda.ucar.edu/datasets/ds313.3/?hash=access (last access: 1 September 2024) (ACOM/CGD, 2018). The SABER data employed in this study are available at https://saber.gats-inc.com/data.php (last access: 1 September 2024). For access to the SD-WACCM-X simulation data, please contact the corresponding authors: Sheng-Yang Gu (gushengyang@whu.edu.cn) or Yusong Qin (qinyusong@whu.edu.cn).

**Supplement.** The supplement related to this article is available online at [the link will be implemented upon publication].

**Author contributions.** Conceptualization and investigation were conducted by JH. Formal analysis and visualization were performed by JH with guidance and supervision from SYG, YSQ, and YFW. Data curation for SD-WACCM-X was carried out by YXL. All authors contributed to the discussion of results and the revision of the manuscript.

**Competing interests.** The contact author has declared that none of the authors has any competing interests.

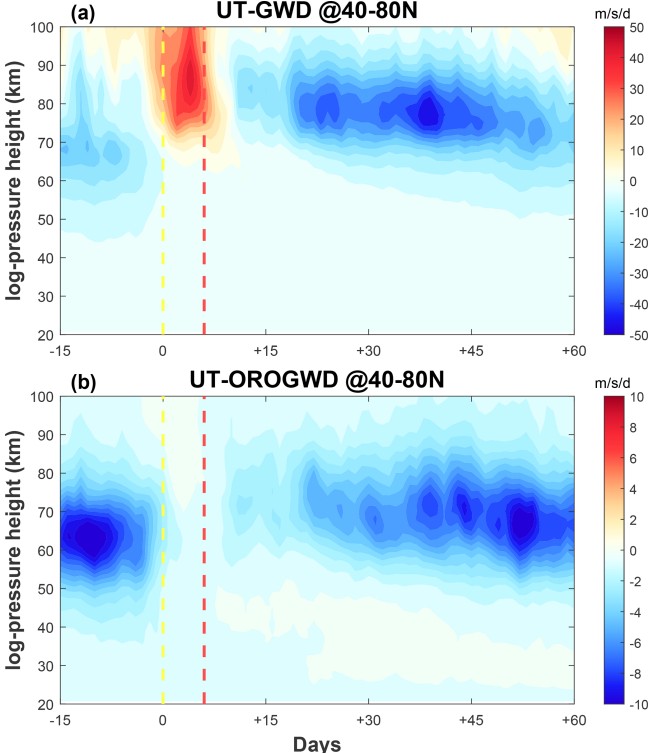

**Figure 10.** Time versus log-pressure height cross-section of SD-WACCM-X composite zonal mean **(a)** zonal wind tendency due to total gravity wave drag (UT-GWD) and **(b)** zonal wind tendency due to orographic gravity wave drag (UT-OROGWD).

**Disclaimer.** Publisher's note: Copernicus Publications remains neutral with regard to jurisdictional claims made in the text, published maps, institutional affiliations, or any other geographical representation in this paper. While Copernicus Publications makes every effort to include appropriate place names, the final responsibility lies with the authors. Views expressed in the text are those of the authors and do not necessarily reflect the views of the publisher.

**Acknowledgements.** The authors acknowledge the National Center for Atmospheric Research (NCAR) for providing the source code of WACCM-X model and NASA for supplying the MERRA-2 reanalysis data and TIMED/SABER OH emission data.

**Financial support.** This research was supported by the National Natural Science Foundation of China (Grant Numbers 42374195 and 42404168), the fellowship of China National Postdoctoral Program for Innovative Talents (Grant Number BX20230273), the Hubei Provincial Natural Science Foundation of China (Grant Number 2024AFB097), and the Postdoctor Project of Hubei Province (Grant Number 2024HBBHCXA054).

**Review statement.** This paper was edited by John Plane and reviewed by Jiarong Zhang and one anonymous referee.

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

## Remarks from the typesetter