# Peer review of "Influence of Sudden Stratospheric Warming With Elevated"

_EGUsphere, 2025_

## Author Comment (AC1)

We are very grateful to the two reviewers for the constructive comments on our manuscript, which greatly helped us to improve the quality of this manuscript. We have now revised the manuscript following your comments and suggestions. Our responses to your comments are listed below in red.

Reviewer #1

I have reviewed the manuscript 'Influence of Major Sudden Stratospheric Warming With Elevated Stratopause on the Hydroxyl in the Polar Middle Atmosphere' by Hu et al.. The authors present the composite response of the polar OH layer in the MLT to 10 ES-SSW events during 2004-2023 using SD-WACCM-X. During the stratospheric warming phase, the peak height of OH layer undergoes a distinct upward displacement, which is closely synchronized with changes in mesospheric temperature, atomic oxygen concentrations, and the vertical component of the residual circulation in the MLT region. GWs play a pivotal role, as the enhanced downward (upward) motion driven by GWs leads to mesospheric warming (cooling) and a corresponding increase (decrease) in atomic oxygen, which in turn facilitates an increase (decrease) in OH concentration. The manuscript is well-written, and the methodology is solid. I have a few comments. My only concern is that this study is mainly based on model results. Please see my major comment for details.

We thank the reviewer for the comment. In the revised manuscript, SABER OH observations for the 2009 SSW event have been added. The observed OH variations show a decrease in peak value and an upward shift, consistent with SD-WACCM-X simulations, which reinforces the robustness of our conclusions.

Major comment:

1.WACCM model has deficiencies in its treatment of GW forcing, which could lead to an underestimation of OH variation during ES-SSWs. I suggest that the authors validate the model results using SABER OH observations—perhaps for at least one SSW event. *Response:* Thank you for your suggestion. Following your advice, we have compared the SD-WACCM-X with SABER OH observations for the 2009 SSW event, as shown in Figure R1d. The SABER results reveal a significant decrease in peak value and an upward shift of the OH layer during the stratospheric warming phase, which is consistent with the SD-WACCM-X simulation. This Figure is added in the revised manuscript.

It should be noted that the SABER OH airglow measurements originate from radiative emissions of vibrationally excited OH (v>0), whereas the SD-WACCM-X output reflects the OH concentration dominated by ground-state OH (v=0). Despite this difference, the simulated OH distribution from SD-WACCM-X exhibits a similar pattern to the SABER observation, because both share the same primary production mechanism. Moreover, the observed peak altitude of OH airglow (~87 km) lies slightly

above the simulated ground-state OH peak (~82km), which is consistent with known collisional quenching processes affecting vibrational levels. This result is added in the revised manuscript (see lines 265-271).

[Figure]

**Figure R1.** Time-altitude cross-section of SD-WACCM-X zonal-mean (a) temperature at 80°N, (b) zonal wind at 60°N, (c) OH concentration (in units of $10^{-9}$), and SABER (d) OH emission during January 06-March 22, 2009. The solid pink line represents the temperature difference between 60°N and 90°N $(T[80° - 90°] - T[60° - 70°])$, and the dashed green line is the stratopause height. The black solid contour in (b) denotes the zero-wind line. Vertical dashed yellow and red lines are the onset of the stratosphere warming stage and the elevated stratopause stage, respectively.

Minor Comments:

1) Lines 190-192: 'In Figure 1a, the meridional temperature gradient (T[80−90]−T[60-70] K) is denoted by the pink solid line, and the height of the ES is indicated by the green dashed line.' How is the height of the ES defined? A sentence in the manuscript would be helpful.

*Response:* Sorry for the confusion. In the original manuscript, the term "height of the ES" was incorrectly used. In Figure 1a, the green dashed line indicates the stratopause height, which is defined as the altitude of maximum temperature within the 20–100 km vertical domain (Chandran et al., 2013). We correct it in the revised manuscript (see lines 221-224).

**Reference:**

Chandran, A., Collins, R. L., Garcia, R. R., Marsh, D. R., Harvey, V. L., Yue, J., and de la Torre, L.: A climatology of elevated stratopause events in the whole atmosphere community climate model, Journal of Geophysical Research-Atmospheres, 118, 1234–1246, https://doi.org/10.1002/jgrd.50123, 2013.

2) Figures 7-9 show latitude-altitude cross-sections of composite relative variation. How is the relative variation defined?

*Response:* Sorry for the confusion. The relative variation in this study is derived from this formula:

$$\text{Reletive variation} = \frac{X_{(i)} - \overline{X}}{\overline{X}} \times 100\% \qquad (1)$$

in which $X_{(i)}$ represents the variable (OH, atomic oxygen, and temperature) for each ES-SSW event, and $\overline{X}$ stands for the background average obtained by averaging the same calendar dates over 2004–2023. The associated statements have been added to the revised manuscript (see lines 446-451).

---

## Author Comment (AC2)

We are very grateful to the two reviewers for the constructive comments on our manuscript, which greatly helped us to improve the quality of this manuscript. We have now revised the manuscript following your comments and suggestions. Our responses to your comments are listed below in red.

Reviewer #2

In the current manuscript, Hu et al have investigated the impact of major sudden stratospheric warming events in the northern hemisphere high latitude on the OH changes in the mesosphere and lower thermosphere. Their work is mainly focusing on the model results (a specific-dynamic version of WACCM-X, SD-WACCM-X) with detailed middle-atmosphere chemistry and D-region chemistry (MAD). They used the MERRA2 reanalysis data to constrain the modelled atmospheric temperature/winds below 50 km from 2004-2023, which can be well reproduce the major SSW events over this period. The authors have made clear introduction and why this work is so important for the atmospheric research community.

First, they choose SSW event in January 2009 to start with and confirmed the elevated stratopause (ES_ and associated temperature/winds/OH changes (which has been studied/revealed by other studies before) in SD-WACCM-X results. Then they found the sharp OH changes are well correlated with meriodinal temperature gradient for the ten SSW events over the past 20 years (2004-2023). Figure 3 shows the climatological OH seasonal variations in the polar regions, which is nothing new here (this seems to be consistent with WACCM and other global atmospheric chemistry models, except the authors found some large discrepancies between SD-WACCM-X and other models). Then the authors have made composite analysis of temperature, zonal mean wind, OH and its anomaly during the ES-SSW events (before 15 days and after 2 months) to see the general pattern (and they thought it does not matter if SSW occurred in December, January or February, their Table1). This Figure 4 may cause some cautions because we can still see there are still some large daily variation in their Figures 1-3. Then they did similar for the atomic oxygen and temperature (Figure 5) to link the OH changes through the chemistry since OH can be depleted by reacting with O in the MLT region. Figures 6-9 shows the OH, O, T and residual circulation for different ES-SSW stages. Finally the authors investigated the causes by the gravity waves.

Overall, the manuscript has very clear message and it reads well. The results and discussions are useful. Personally I enjoy reading it. However, I am not sure if the conclusion would be changed if their current method changes (including the comments mentioned above).

We thank the reviewer for the positive comments and for raising this important point. We agree that daily variability is still visible in Figures 1–3. However, the purpose of the composite analysis in Figure 4 is to extract the robust and systematic signals associated with ES-SSW events from individual event variability. As shown in Figure 4d, the OH anomalies exhibit a coherent response to ES-SSW events, which cannot be explained by daily variations alone.

It should also be noted that Figure 3 is included to characterize the seasonal variability of OH in the polar regions. Although the general seasonal pattern has been reported in previous studies, presenting it here based on the same SD-WACCM-X simulations allows us to establish a consistent reference and reduce potential errors when comparing climatology with anomalies. The associated statements have been added to the revised manuscript (see lines 311-323).

We also note that the consistency of the OH response across ten ES-SSW events during 2004–2023 further supports the robustness of the results. Therefore, while daily variability remains present in individual cases, the composite analysis demonstrates that our main conclusions are not sensitive to the specific onset date or to short-term variability, but rather reflect the common dynamical and chemical processes associated with ES-SSW events.

1. I have noticed some differences/inconsistent of major SSW occurring date with published data at: https://csl.noaa.gov/groups/csl8/sswcompendium/majorevents.html although the definition of SSW is slightly different to Charlton and Polvani (2007) (https://journals.ametsoc.org/view/journals/clim/20/3/jcli3996.1.xml). For example, I did not see SSW events on 25 December 2018 (in Table 1) from the table in the above NOAA webpage (which showed it should be 2 January 2019 from MERRA2).

[Figure]

**Figure R1.** Temporal variation of the temperature difference between 60°N and 90°N at 10 hPa (red line) and the zonal-mean zonal wind at 60°N and 10 hPa (blue line). Vertical dashed green and blue lines denote the onset of the positive temperature difference and the zonal wind reversal, respectively.

*Response:* We thank the reviewer for pointing out the inconsistency of the SSW onset between the NOAA and our study. As the reviewer mentioned, the definition of SSW onset can vary among studies. While Charlton and Polvani (2007) and the NOAA typically define the onset based on the reversal of the zonal-mean wind at 60°N and 10 hPa, in our study, we adopt the day when the temperature difference turns positive as the onset date. Under this definition, we determine the onset on 25 December 2018.

As illustrated in Figure R1, the red line shows the meridional temperature difference from SD-WACCM-X, which turns positive on 25 December 2018. Based on our definition, this date is identified as the onset of the SSW. In contrast, the blue line represents the zonal-mean zonal wind at 60°N and 10 hPa from MERRA2, which reverses from westerlies to easterlies on 2 January 2019, consistent with the NOAA definition. The difference in onset timing, therefore, arises solely from the diagnostic criterion employed, rather than from inconsistencies in the data or simulation.

It should be noted that although the onset definition differs, previous studies also indicated that the 2019 SSW exhibited clear dynamical and thermal responses in late December 2018 (e.g., Gan et al., 2020; Ma et al., 2020; Yamazaki et al., 2025). In addition, our OH results are consistent with earlier findings (e.g., Damiani et al., 2010), further supporting the robustness of the adopted criterion. Therefore, the difference from the NOAA record (2 January 2019 from MERRA2) arises from diagnostic criteria rather than a mistake in our analysis, and it does not affect the subsequent results and conclusions of this study.

**Reference:**

Damiani, A., Storini, M., Santee, M. L., and Wang, S.: Variability of the nighttime OH layer and mesospheric ozone at high latitudes during northern winter: influence of meteorology, Atmospheric Chemistry and Physics, 10, 10291–10303, https://doi.org/10.5194/acp-10-10291-2010, 2010.

Gan, Q., Eastes, R. W., Burns, A. G., Wang, W., Qian, L., Solomon, S. C., Codrescu, M. V., and McClintock, W. E.: New observations of large-scale waves coupling with the ionosphere made by the GOLD Mission: Quasi-16-day wave signatures in the F-region OI 135.6-nm nightglow during sudden stratospheric warmings, Journal of Geophysical Research: Space Physics, 125, e2020JA027880, 2020.

Ma, Z., Gong, Y., Zhang, S., Zhou, Q., Huang, C., Huang, K., Luo, J., Yu, Y., and Li, G.: Study of a Quasi 4-Day Oscillation During the 2018/2019 SSW Over Mohe, China, JOURNAL OF GEOPHYSICAL RESEARCH-SPACE PHYSICS, 125, https://doi.org/10.1029/2019JA027687, 2020.

Yamazaki, Y., Sato, K., Koshin, D., and Yasui, R.: Atmospheric semidiurnal solar tide response to sudden stratospheric warmings in the JAGUAR-DAS Whole neutral Atmosphere Reanalysis (JAWARA) during 2004–2023, Journal of Geophysical Research: Space Physics, 130, e2024JA033688, 2025.

2.    I expect ES-SSW onset date is later than SSW occurring date since the breaking of planetary waves propagate upward. However, the Table 1 has earlier onset ES-SSW date than SSW. Not sure if this caused by SD-WACCM-X simulation or the authors made some mistakes. If the basic onset day in Table 1 is wrong, obviously, all other figures (except Figure 3), results and discussion need to be changed.

*Response:* We thank the reviewer for the careful observation. We would like to clarify that an ES-SSW is a specific subtype of SSW. The elevated stratopause (ES) appears after the stratospheric warming has taken place; therefore, the occurrence of ES-SSW is inherently synchronized with that of the SSW, and its onset is the same as the SSW onset. The apparent difference in onset dates arises only from the choice of diagnostic criterion. In this study, we adopt the day when the zonal-mean temperature difference turns positive as the onset of SSW (and thus ES-SSW). By this definition, the 2019 event onset is determined to be 25 December 2018. In contrast, the NOAA and some other studies use the reversal of zonal-mean zonal wind at 60°N and 10 hPa, which gives 2 January 2019 as the onset.

Therefore, the earlier onset date listed in Table 1 is not due to mistakes in the SD-WACCM-X simulation or in our analysis, but rather reflects the use of a different diagnostic definition. Importantly, this choice does not affect the identification of ES-SSW events or the robustness of the subsequent results and conclusions.

3. For the residual circulation, why SD-WACCM looks different to WACCM (for example, Anne Smith's Figure at JGR(2011): please see the link: https://agupubs.onlinelibrary.wiley.com/doi/full/10.1029/2011JD016083. Maybe this is from WACCM6 physics and it is quite different to WACCM4 using in Smith et al. (2011). Not sure why the authors choose the most expensive SD-WACCM6-X (MAD) model (FXSD) since their work is mainly focusing in the MLT below 95 km? It seems to me this is the first long term SD-WACCM6-X simulation, it is unclear if this is from cesm2_1_3 or other version. So additional model validation is required.

[Figure]

**Figure R2.** Vectors showing the WACCM climatological TEM flow for December averaged over 1960–2006 for four model realizations. The scale at the lower right shows the maximum equation image* and equation image* in m/s. For reference, the right axis on all figures gives the climatological global mean geometric altitude from WACCM (Smith et al., 2011).

[Figure]

**Figure R3.** Vectors showing the SD-WACCM-X climatological residual circulation for December averaged over 2004–2023.

*Response:* We are grateful to the reviewer for the thoughtful comparison of our results with those of Smith et al. (2011). The apparent discrepancy arose because our original manuscript mistakenly described Figures 7-9 as climatological residual circulation, while they show the residual circulation anomaly. We have corrected this wording in the revised manuscript to avoid confusion.

To further clarify, we compared the climatological residual circulation reported by Smith et al. (2011) with the December climatology derived from SD-WACCM-X simulations (2004–2023 average). Their close agreement (Figures R2-R3) confirms the reliability and accuracy of SD-WACCM-X.

Regarding the choice of model, we employed SD-WACCM6-X because it is a whole-atmosphere model that self-consistently couples the middle atmosphere, thermosphere, and ionosphere. Although the present study focuses on the MLT region below 95 km, we performed the simulations over the entire atmospheric column since the outputs above 95 km will be used in our future studies. Running SD-WACCM6-X provides a consistent dataset that can be applied not only to the present analysis of OH variations but also to upcoming investigations of upper atmospheric and ionospheric processes.

The simulations were performed with CESM2.2.0, and the model description can be found in Gettelman et al. (2019). The associated statements have been added to the revised manuscript (see lines 119-120).

**Reference:**

Gettelman, A., Mills, M. J., Kinnison, D. E., Garcia, R. R., Smith, A. K., Marsh, D. R., Tilmes, S., Vitt, F., Bardeen, C. G., McInerny, J., Liu, H. L., Solomon, S. C., Polvani, L. M., Emmons, L. K., Lamarque, J. F., Richter, J. H., Glanville, A. S., Bacmeister, J. T., Phillips, A. S., Neale, R. B., Simpson, I. R., DuVivier, A. K.,

Hodzic, A., and Randel, W. J.: The Whole Atmosphere Community Climate Model
Version 6 (WACCM6), Journal of Geophysical Research-Atmospheres, 124,
12380–12403, https://doi.org/10.1029/2019jd030943, 2019.

Smith, A. K., Garcia, R. R., Marsh, D. R., and Richter, J. H.: WACCM simulations of
the mean circulation and trace species transport in the winter mesosphere, Journal
of Geophysical Research: Atmospheres, 116, 2011.

4.  Even if this is from cesm2_1_3, what the inputdata (emissions etc) are used beyond
January 2015? It requires some clarification and make major changes before moving
forward.

[Figure]

**Figure R4.** Temporal variation of OH concentrations in the polar region. Panels from
left to right correspond to 2009, 2010, and 2013, and from top to bottom to SD-
WACCM-X and SD-WACCM-X (No Specifier) simulations. Vertical dashed yellow
and red lines are the onset of the stratosphere warming stage and the elevated
stratopause stage, respectively.

*Response:* We thank the reviewer for raising this important point. To the best of our
knowledge, CESM currently provides emission input files for the FXSD component
only up to the year 2015. Accordingly, in our simulations, prescribed emission data files
from        CESM        input        (available        at        https://svn-ccsm-
inputdata.cgd.ucar.edu/trunk/inputdata/atm/cam/chem/emis/CMIP6_emissions_1750_
2015_2deg) were used prior to 2015, while for years beyond 2015, the emission fields
were set to missing values (by specifying in the user_nl_cam file:

    ext_frc_type    = 'CYCLICAL',
    ext_frc_specifier    = ' ',

```
srf_emis_type = 'CYCLICAL',
srf_emis_specifier = ' ').
```

To evaluate the potential impact of this treatment, we conducted sensitivity tests for three years before 2015 (2009, 2010, and 2013). For each year, simulations were performed both with the prescribed emissions and with the emission fields set to NaN. The results (see Figure R4) show that the temporal evolution of OH is nearly identical between the two cases. For example, in the 2009 event, both simulations reproduce a pronounced decrease in OH peak concentration during the stratospheric warming phase. Although some differences in the magnitude of OH variations can be noted between the two cases, the dynamical processes and response characteristics of the SSW remain unaffected. Moreover, comparisons across years confirm that SD-WACCM-X results are consistent regardless of whether emissions are specified.

These sensitivity tests demonstrate that the absence of emission data after 2015 has a negligible impact on the OH response to SSWs in the polar region. Therefore, we are confident that the results presented in this study remain robust and reliable. The associated statements have been added to the revised manuscript (see lines 145-153).

---

## Author Response (AR2)

We are very grateful to the two reviewers for the constructive comments on our manuscript, which greatly helped us to improve the quality of this manuscript. We have now revised the manuscript following your comments and suggestions. Our responses to your comments are listed below in red.

Reviewer #1

1. In the response letter, the authors provided a link to the prescribed emission data files (https://svnccsminputdata.cgd.ucar.edu/trunk/inputdata/atm/cam/chem/emis/CMIP6_emissions_1750_2015_2deg). However, the link is not accessible.

*Response:* We apologize for the confusion. The correct link is https://svn-ccsm-inputdata.cgd.ucar.edu/trunk/inputdata/atm/cam/chem/emis/CMIP6_emissions_1750_2015_2deg. Please note that the hyphen in "svn-ccsm-inputdata" is part of the official URL and not a typographical error.

We are very grateful to the two reviewers for the constructive comments on our manuscript, which greatly helped us to improve the quality of this manuscript. We have now revised the manuscript following your comments and suggestions. Our responses to your comments are listed below in red.

Reviewer #2

The authors have sufficiently addressed my questions. Here are some minor comments to improve the paper:

1. It would be good to provide the definition of elevated stratospause event due to the major sudden stratospheric warming. Please note that the ES event does not always happen after SSW. I would be even better to add some mechanism about ES occurrences after SSW (in the introduction section after Lines 84-86) based on previous studies.                                    For                                    example: https://agupubs.onlinelibrary.wiley.com/doi/full/10.1029/2011JD016840#

*Response:* We thank the reviewer for this helpful suggestion. The new text "Elevated stratopause (ES) events refer to episodes during which the winter polar stratopause initially descends, subsequently becomes indistinct, and eventually reforms at a significantly higher altitude (Manney et al., 2008). Such events arise from strong forcing of the zonal wind and meridional circulation by planetary waves (Torre et al., 2012). In certain cases, ES events occur in connection with SSW events. According to Chandran et al. (2013), ES-SSW events are distinguished by a prolonged reversal of the stratospheric jet, enhanced gravity wave forcing, and intensified mean meridional circulation, relative to winters when SSWs occur without an accompanying ES. Compared with typical SSWs, ES-SSW events are particularly noteworthy because their enhanced downward transport significantly modifies the concentration of minor species in the MLT region (Chandran et al. 2013), such as OH. Given these persistent anomalies, focusing on ES-SSW events provides a physically meaningful basis for composite analysis." has been inserted in the introduction (see lines 84–97).

2. In the line 152, it would be better to add "mesosphere" before OH. In factor, other altitude range for example in the lower stratosphere/troposphere OH will change if the emissions during 2015-2023 are set to zero below 15 km.

*Response:* We thank the reviewer for this helpful suggestion. We have revised the text to specify 'mesospheric OH' in the revised manuscript (see line 158).

3. Line 197-198: I am so surprised that it uses 20-100 km range for the stratosphere though we know what the temperature structure looks like. This must be narrowed down quite lot. The averaged stratospause altitutde is around 50-60 km and the elevated ES altitude is around 80 km. Not sure if any of stratopause altitude is below 45 km or above 90 km...

*Response:* We appreciate the reviewer's comment. This height range (20–100 km) is intentionally retained in the analysis because it effectively avoids potential truncation of the temperature maximum during ES-SSW events and ensures full coverage of both

the descending and re-forming stratopause. For instance, during the 2009 SSW, the stratopause altitude descended to as low as ~37 km in the stratospheric warming phase, which justifies the need for an extended vertical range to accurately capture the complete evolution of the stratopause height. Similar vertical domains have also been adopted in previous studies (e.g., Chandran et al., 2013; Torre et al., 2012).

**Reference:**

Chandran, A., Collins, R. L., Garcia, R. R., Marsh, D. R., Harvey, V. L., Yue, J., and de la Torre, L.: A climatology of elevated stratopause events in the whole atmosphere community climate model, Journal of Geophysical Research-Atmospheres, 118, 1234–1246, https://doi.org/10.1002/jgrd.50123, 2013.

de la Torre, L., Garcia, R., Barriopedro, D., and Chandran, A.: Climatology and characteristics of stratospheric sudden warmings in the Whole Atmosphere Community Climate Model, JOURNAL OF GEOPHYSICAL RESEARCH-ATMOSPHERES, 117, https://doi.org/10.1029/2011JD016840, 2012.

4. Unit: Sometimes the current version uses "in the unit of 10^-9" for OH concentration. It would be better to use a proper unit for example: volume mixing ratio (vmr). I thought the SABER OH emission unit is probably wrong (Fig1d), which should be ergs/cm3/sec, please double check it.

*Response:* We thank the reviewer for pointing this out. The OH concentration has been uniformly expressed in units of parts per billion by volume (ppbv) throughout the revised manuscript. In addition, the unit of the SABER OH emission shown in Figure 1d has been corrected in the revision.

5. I am not convinced by the explanation in Lines 265-271 becuase you are comparing two different things (one is the distribution of OH gas, the other is airglow emissions which is determined by the H+O3 reaction and can be largely affected by mesosphere dynamics (e.g., Plane et al., 2015, DOI: 10.1021/cr500501m), some of them have also discussed in the Section 4.1

*Response:* We thank the reviewer for this insightful comment. According to Damiani et al., (2010), OH in the mesopause region is primarily produced through the reaction between ozone and atomic hydrogen, forming excited OH* near 87 km, which constitutes the well-known mesospheric nightglow layer. This excited state is deactivated either via photon emission in the Meinel bands (observed as nightglow) or by collisional quenching. The latter process prevails at lower altitudes, where higher atmospheric density facilitates the formation of a ground-state OH layer near 82 km. Although the SABER OH emission and SD-WACCM-X OH concentration represent physically distinct quantities, both are governed by closely coupled mesospheric dynamical and chemical processes. For example, Gao et al., (2011) and Winick et al., (2009) demonstrated that variation in OH emission during the 2009 SSW were modulated by changes in atomic oxygen and mesospheric temperature, and Damiani et al., (2010) shown that the displacement of OH layer is coupled to variations in oddoxygen (Ox) and mesospheric temperature. These findings indicate that the variations of OH emission and ground-state OH concentration are dynamically linked.

In this study, we focus only on comparing the temporal evolution of OH during SSW events, rather than performing a quantitative comparison of their absolute magnitudes. Accordingly, a qualitative comparison between the two is physically meaningful. This clarification has been incorporated into the revised manuscript (see lines 284-297).

**Reference:**

Damiani, A., Storini, M., Santee, M. L., and Wang, S.: Variability of the nighttime OH layer and mesospheric ozone at high latitudes during northern winter: influence of meteorology, Atmospheric Chemistry and Physics, 10, 10291–10303, https://doi.org/10.5194/acp-10-10291-2010, 2010.

Gao, H., Xu, J. Y., Ward, W., and Smith, A. K.: Temporal evolution of nightglow emission responses to SSW events observed by TIMED/SABER, Journal of Geophysical Research-Atmospheres, 116, https://doi.org/10.1029/2011jd015936, 2011.

Winick, J., Wintersteiner, P., Picard, R., Esplin, D., Mlynczak, M., Russell, J., and Gordley, L.: OH layer characteristics during unusual boreal winters of 2004 and 2006, JOURNAL OF GEOPHYSICAL RESEARCH-SPACE PHYSICS, 114, https://doi.org/10.1029/2008JA013688, 2009.

6. The other very technique question about the Y-axis in all the figures regarding to "Altitude (km)". Have you interpolated the modelled goepotential heigt to fixed altitude, or you use some approximate relationship between pressure and height, or use the some equation to calculate the altitude. This is also important since the layer will move a little bit up and down depending on your method.

***Response:*** We thank the reviewer for this technical question. In this study, the model pressure levels are converted to approximate geometric altitude using the standard hydrostatic relationship ($Z = -H \times \ln(P/P_s)$) (Andrews et al., 1987).

We compared the OH distributions derived from (a) log-pressure–based altitude and (b) model geopotential height interpolated onto fixed geometric altitude levels (see Figure R1). The two methods exhibit noticeable differences in the peak altitude of the OH layer. For example, after January 26, 2009, the OH layer peak in log-pressure height (Figure R1a) descends to ~78 km, whereas in geometric coordinates the descent is even larger (Figure R1b), reaching ~74 km. These phenomena are consistent with OH emission shown in Grygalashvyly et al. (2014), who showed that geometric altitudes are affected by shrinking of middle atmosphere (SMA), while pressure coordinates remain largely free of this effect. As a result, the OH peak expressed in log-pressure height tends to appear somewhat higher than when expressed in geometric height.

Previous studies (e.g., Pickett et al., 2006) demonstrated that the OH layer is markedly more stable in pressure coordinates than in geometric altitude. Meanwhile, as noted by Grygalashvyly et al. (2014), pressure coordinates are more natural for

discussing chemical, physical, and radiative processes, as these processes are largely free of the SMA effects on pressure isosurfaces. Guided by these findings, we use log-pressure height ( $Z = -H \times \ln(P/P_s)$ ) as the vertical coordinate for diagnosing the OH responses in SD-WACCM-X, while also providing an approximate geometric-height scale for reference. This clarification has been incorporated into the revised manuscript (see lines 221-229).

[Figure]

**Figure R1.** Comparison of the OH variations during the 2009 SSW event derived from (a) log-pressure–based altitude and (b) model geopotential height interpolated onto fixed altitude levels.

**Reference:**

Andrews, D. G., Leovy, C. B., and Holton, J. R.: Middle atmosphere dynamics, Academic press1987.

Grygalashvyly, M., Sonnemann, G., Lübken, F. J., Hartogh, P., and Berger, U.: Hydroxyl layer: Mean state and trends at midlatitudes, Journal of Geophysical Research: Atmospheres, 119, 12,391–312,419, https://doi.org/10.1002/2014JD022094, 2014.

Pickett, H., Read, W., Lee, K., and Yung, Y.: Observation of night OH in the mesosphere, Geophysical Research Letters, 33, https://doi.org/10.1029/2006GL026910, 2006.